# Wearable Woven Triboelectric Nanogenerator Utilizing Electrospun PVDF Nanofibers for Mechanical Energy Harvesting

**DOI:** 10.3390/mi10070438

**Published:** 2019-06-30

**Authors:** Muhammad Omar Shaikh, Yu-Bin Huang, Cheng-Chien Wang, Cheng-Hsin Chuang

**Affiliations:** 1Institute of Medical Science and Technology, National Sun Yat-sen University, Kaohsiung 80424, Taiwan; 2Department of Mechanical Engineering, Southern Taiwan University of Science and Technology, Tainan 71005, Taiwan; 3Department of Chemical and Materials Engineering, Southern Taiwan University of Science and Technology, Tainan 71005, Taiwan

**Keywords:** triboelectric nanogenerator, energy harvesting, human motion, self-powered wearables, electrospinning, polyvinylidene fluoride

## Abstract

Several wearable devices have already been commercialized and are likely to open up a new life pattern for consumers. However, the limited energy capacity and lifetime have made batteries the bottleneck in wearable technology. Thus, there have been growing efforts in the area of self-powered wearables that harvest ambient mechanical energy directly from surroundings. Herein, we demonstrate a woven triboelectric nanogenerator (WTENG) utilizing electrospun Polyvinylidene fluoride (PVDF) nanofibers and commercial nylon cloth to effectively harvest mechanical energy from human motion. The PVDF nanofibers were fabricated using a highly scalable multi-nozzle far-field centrifugal electrospinning protocol. We have also doped the PVDF nanofibers with small amounts of multi-walled carbon nanotubes (MWCNT) to improve their triboelectric performance by facilitating the growth of crystalline β-phase with a high net dipole moment that results in enhanced surface charge density during contact electrification. The electrical output of the WTENG was characterized under a range of applied forces and frequencies. The WTENG can be triggered by various free-standing triboelectric layers and reaches a high output voltage and current of about 14 V and 0.7 µA, respectively, for the size dimensions 6 × 6 cm. To demonstrate the potential applications and feasibility for harvesting energy from human motion, we have integrated the WTENG into human clothing and as a floor mat (or potential energy generating shoe). The proposed triboelectric nanogenerator (TENG) shows promise for a range of power generation applications and self-powered wearable devices.

## 1. Introduction

In recent years, there has been a shift towards “wearable electronics” to meet the requirements of modern living such as motion tracking, physical health monitoring and artificial skin sensors among others [1,2,3,4,5,6,7]. These wearable electronics are likely to open up new applications and can be integrated with currently existing wearables like wristwatches [8] and eyeglasses [9]. One common drawback of these wearables is that they all need to be powered by rechargeable batteries, which limit their lifetime and sustainability [10,11]. Furthermore, it almost becomes impossible to replace or recharge these batteries due to their distributed nature over multiple devices. Thus, there have been growing efforts to make these wearable electronics “self-powered” and sustainable by directly harnessing energy from human motion. Since human activity is based on mechanical movement, harnessing this ubiquitous source of biomechanical energy represents the most reliable strategy to generate power for wearables [12,13,14].

To date, several approaches have been proposed to harvest mechanical energy and convert it into useful electrical energy, which includes electrostatic [15], piezoelectric [16] and triboelectric [17] among others. Among the various approaches, triboelectric nanogenerators or triboelectric nanogenerators (TENGs) have gained significant research interest due to merits like a wide choice of materials and nanogenerator designs, low cost, high efficiency, robustness, flexibility and being environmentally friendly [15,16,17,18,19,20,21,22]. TENGs can harvest energy from ambient mechanical motion like vibration, rotation, expansion and contraction and are based on contact electrification and electrostatic induction arising from the friction of two materials that have different affinities for electrons [23,24]. Several materials have been employed as dissimilar triboelectric surfaces and a larger difference in the electron affinity of the two materials results in higher output performance of the TENG. Until now, a range of applications have been successfully demonstrated for TENGs based on the relative position change between two dissimilar triboelectric materials, such as shoe insole to harvest energy from human walking [25], smart clothing for harvesting energy from human motion [26] and harvesting high altitude wind energy [27] among others. The output performance of TENGs has improved significantly in recent years owing to novel designs and materials optimization. However, while TENGs based on micro and nanopatterned silicone and other polymeric films have demonstrated energy harvesting capability for a range of applications, they are not ideal materials for direct integration with regular clothing due to issues associated with washability and air permeability. Consequently, there has been a growing trend to develop smart textiles based on fibers that provide excellent deformability, breathability and washability while enabling easy and direct incorporation into everyday clothing [28,29,30].

Herein, we have developed a woven structured TENG (WTENG) based on freestanding triboelectric mode utilizing polyvinylidene fluoride (PVDF) nanofibers (as the electronegative material) woven together with a commercial nylon cloth (as the electropositive material) to harvest energy from human motion. PVDF is an attractive non-reactive polymeric material with unique electroactive properties, low acoustic impedance and flexibility, thus making it an effective dielectric for triboelectric applications [31]. The PVDF nanofibers are fabricated using a low cost and scalable centrifugal electrospinning process and have small structural features that significantly increase the surface area on contact, thus resulting in an amplified response. Furthermore, these fibers have also been doped with varying concentrations of multi-walled carbon nanotubes (MWCNT) which have shown to increase the overall triboelectric charge generation. The proposed WTENG does not utilize any nanostructured patterning, thus resulting in increased reliability over its lifetime and ease of fabrication. At first, the working principle is analyzed followed by testing it under different contact conditions to observe the electrical output. Finally, the WTENG is applied to harvest different forms of mechanical energy from human motion such as movement of arms and walking.

## 2. Materials and Methods

### 2.1. Fabrication of PVDF Nanofibers

The PVDF nanofibers were fabricated using a far-field centrifugal electrospinning (CE) protocol where the synergistic effect of the centrifugal drawing force and the conventional electrical force results in nanofibers with improved yields [32]. The schematic of the CE setup is shown in Figure 1 and the aim was to enable scalable mass manufacturing of nanofibers. First, the spinning solution was prepared by adding 7 wt. % semi crystalline PVDF powder (*M_w_* ~534,000 by GPC, Sigma-Aldrich, St. Louis, MO, USA) in a solution of Dimethylformamide (DMF) and Tetrahydrofuran (THF) where the ratio of DMF to THF was 39:1. The PVDF powder was completely dissolved by heating the solution at 50 °C for 1 h under constant magnetic stirring. The addition of THF reduces the boiling point of the solution, which aids in evaporation during electrospinning, thus resulting in improved fiber formation. The CE setup consists of an inner Teflon tubular column attached to a speed adjustable three phase motor and an outer cylindrical stainless-steel collector, with a diameter of 40 cm, which was coated on the inside with an aluminum foil. The prepared spinning solution enters the inner tube at a constant flow rate of about 45 drops/min. Using a slower flow rate causes the droplets to be pre-polarized in the feeding tube and attach to its inner walls. The bottom of the inner tube has a diameter of 15 cm with twelve metal nozzles, each with a diameter of 0.5 mm, through which the solution jet was ejected simultaneously. A high voltage power supply was used to apply a voltage of 50 kV to the metal nozzles through a carbon brush, with resulting electric field strength of 200 kV/m, and the ejected nanofibers were collected onto the grounded aluminum foil. The solution jet trajectory initially follows a straight path and as it moves towards the outer cylindrical collector, a bending instability develops and the repulsive charges on the jet cause it to elongate. Consequently, ultrafine fibers with nanoscale dimensions and high aspect ratios were obtained.

### 2.2. Doping with Multi-Walled Carbon Nanotubes (MWCNT)

The PVDF nanofibers were doped with MWCNT to facilitate the growth of crystalline β-phase and enhance the surface charge density on the nanofibers during contact electrification. Before the doped spinning solution was prepared, the MWCNT were first acid treated to introduce functional groups which can improve their dispersion in the solvent. Briefly, the MWCNT were added to a concentrated H_2_SO_4_/HNO_3_ (1:3 vol %) solution and stirred for 12 h at 60 °C. This was followed by an ultrasonication step for 1 hour to remove impurities and increase the concentration of carboxylic groups on the surface of the MWCNT. Extra acid was removed via filtration and the MWCNT slurry was washed copiously with deionized water. The oxidized MWCNT were dried in a vacuum oven at 75 °C overnight followed by dispersion in DMF:THF (39:1 vol %) solvent solution. The acid treatment resulted in an extremely stable and uniform dispersion of MWCNT in the solvent. Finally, the solvent solutions containing MWCNT and PVDF powder were mixed, where the doping concentration of MWCNT was varied up to 1 wt. %. The doped PVDF nanofibers were obtained by spinning the prepared solution using the same CE protocol as described in the previous section.

### 2.3. Polyvinylidene Fluoride (PVDF) Nanofiber Characterization

The morphology and diameter of the PVDF nanofibers was characterized by a field emission scanning electron microscope (FE-SEM, JSM-6701F, JEOL, Tokyo, Japan). The crystal structure of the nanofibers was determined by X-Ray Diffraction (XRD, Bruker Taiwan Co. Ltd, Hsinchu, Taiwan) where the incident and diffracted X-ray beams were vertical to the nanofiber mat and analysis was performed using a scan range of 0−30° and a scan rate of 0.5 degrees/min.

### 2.4. Woven Triboelectric Nanogenerator (WTENG) Fabrication

The WTENG fabrication primarily consists of preparing the individual PVDF nanofiber and nylon based single electrodes followed by weaving them together. The systematic fabrication protocol is schematically illustrated in Figure 2a. First, an aluminum foil (5 mm × 60 mm) with a soldered copper wire is coated on both sides with a 50 µm thick double-sided tape (Taiwan Nitto Co., Ltd., Taichung, Taiwan). The PVDF based electrode is obtained by attaching the PVDF nanofiber mat to each side of the aluminum foil using thermocompression bonding. The nylon-based electrode is obtained in a similar way by attaching the nylon cloth to both sides of the aluminum foil. Lastly, the prepared PVDF and nylon based single electrodes are woven together to form a 10 × 10 WTENG, which has a total area of about 60 mm^2^, as shown in Figure 2b.

### 2.5. Measurement Setup

We have utilized a standard dynamic testing platform as shown in the schematic in Figure 3 to test the electrical output of the WTENG when mechanically contacted with a freestanding dielectric elastomeric layer (PDMS 184) under different applied forces and frequencies. A function generator (AFG3022, Tektronix Inc., Beaverton, OR, USA) linked to a linear power amplifier (PA25E, Brüel & Kjær Co., Nærum, Denmark) was used to control the excitation frequency and normal force exerted by the shaker, respectively. A force feedback sensor attached to the front of the shaker measures the applied contact force, which is displayed as voltage waveforms on the digital phosphor oscilloscope after signal conditioning. A 50 mm^2^ PDMS layer is attached to the tip of the force sensor as it contacts the surface of the 60 mm^2^ WTENG that is fixed on an acrylic block. The open circuit voltage and the transferred charge were measured using an electrometer (Keithley 6514 System Electrometer, Beaverton, OR, USA) with a high input resistance. For simulating human motion such as walking and arm movement, the WTENG was contacted under different forces and frequencies and the resulting electrical output was measured by the electrometer.

## 3. Results and Discussion

### 3.1. Polyvinylidene Fluoride (PVDF) Nanofibers

The morphology of the pure and MWCNT doped PVDF nanofiber mat obtained via centrifugal electrospinning was observed using FE-SEM, as shown in the images in Figure 4a,b. It can be seen that PVDF nanofibers with diameters ranging from 100 nm to 300 nm were obtained with fully interconnected pores and a high degree of porosity of about 70% to 80%. Furthermore, the fiber surface was smooth without any observable bead formation or presence of wrinkles. The smooth surface and porous structure of the nanofibers can be attributed to the DMF: THF solvent ratio used during electrospinning. The use of a cosolvent with high DMF content results in a smooth fiber surface while the use of THF that has a low boiling point and high vapor pressure increases the porosity of the fibers.

Among the crystalline phases present in PVDF, the non-polar α-phase consisting of TGTG (alternating trans gauche confirmation) is the most abundant form while the β-phase consisting of TTTT (all trans confirmations) is polar and demonstrates the largest spontaneous polarization per unit cell. Herein, we have tried to use a combination of electrospinning and MWCNT doping to synergistically increase the β-phase content in the obtained PVDF nanofibers. The electrospinning process results in uniaxial mechanical drawing and electrical poling of the PVDF nanofibers. The mechanical drawing process forces the molecules into an extended conformation with all dipoles aligned in the same direction (β-phase). Simultaneously, the poling process causes the polar axis of the crystallites to be oriented along the electric field direction, resulting in spontaneous polarization of the β-phase. In addition, doping with MWCNT further promotes the conversion of the α-phase into the β-phase by increasing the crystallization rate and inducing interfacial charge accumulation. This increased polarization and charge storage ability of the PVDF nanofibers due to MWCNT doping should result in an increased triboelectric output performance of the WTENG.

PVDF nanofibers with high β-phase content can be obtained by cooperative effect of the electrospinning process and addition of small amounts of MWCNT as confirmed by the XRD spectra shown in Figure 4c. The as obtained PVDF powder showed strong absorption peaks at 18.6°, 20.1° and 26.8° which corresponds to the (020), (110) and (021) planes of the α-phase. In the case of the bare PVDF nanofiber mat obtained after centrifugal electrospinning, the peaks related to the α-phase decreased significantly and a new peak at 20.9° emerged which corresponds to the (110) plane of the β phase. Furthermore, this peak became more dominant as the PVDF nanofibers were doped with increasing concentrations of MWCNT from 0.5 to 1 wt. %. We have utilized a 1 wt. % MWCNT loading to fabricate the W-TENG as the relative peak intensity corresponding to the β phase decreases for higher doping concentrations.

To analyze the triboelectric response, a single electrode system comprising of electrospun PVDF nanofiber mat (with and without MWCNT) attached to an aluminum foil with dimensions of 25 mm^2^ was fabricated and periodically contacted with a PDMS sheet under a normal applied force of 3 N and a frequency of 1 Hz. The resulting electrical output as measured by the electrometer is shown in Figure 4d. The pure PVDF nanofibers produced an average peak-to-peak current of about 4 nA that increased to about 8 nA and 16 nA as the MWCNT loading increased from 0.5 wt. % to 1 wt. %, respectively. These results highlight the feasibility of the MWCNT loading in enhancing the triboelectric response.

A few recent studies have also been reported in literature utilizing an increase in β-phase content of PVDF to enhance the triboelectric response. Kim et al. [33], who fabricated a TENG with a PVDF-MWCNT nanocomposite film and aluminum tape as the two triboelectric layers, reported similar observations. The nanocomposite film was formed by dropping and drying the composite solution and they observed a maximum output voltage that was about eight times higher than that for the pure PVDF based TENG. These findings lead them to conclude that the increased β-phase content with a highly polarized crystal structure resulted in the enhanced triboelectric response. Recently, Lee et al. synthesized poly (tert-butyl acrylate) (PtBA)-grafted PVDF copolymers and demonstrated that the copolymers were very effective in increasing TENG output performance as compared to pristine PVDF [34]. The grafting process produced high β-phase content that doubled the measured dielectric constant while the output performance of the TENG also showed a two-fold enhancement. They concluded that the high net dipole moment of the β-phase and resulting increase in dielectric constant enhanced the surface charge density during contact electrification, which was responsible for the improved TENG response. This was one of the first studies to experimentally prove the direct correlation between the phase and dielectric constant with the output power of TENG. In addition, Soin et al. [35] reported a triboelectric generator with vertical contact-separation mode that consisted of zinc stannate nanocubes-PVDF nanocomposites and a polyamide-6 membrane. As compared to the pristine PVDF, the nanocomposites showed a higher β-phase content and dielectric constant with an increased piezoelectric coefficient. Furthermore, the resulting triboelectric output voltage and current also showed an enhancement of 70% and 200% respectively. This improved performance was related to the enhanced polarization of PVDF leading to an increase in the β-phase content and higher surface charge density by stress induced polarization of zinc stannate.

### 3.2. Freestanding TENG—Working Mechanism

To understand the working mechanism of the WTENG, we have first analyzed a freestanding TENG (FTENG) using PVDF nanofiber mat (25 mm × 25 mm) and nylon fabric (25 mm × 25 mm) as the two triboelectric surfaces and two aluminum foils as the electrodes. The structure of the FTENG under vertical contact mode is schematically illustrated in Figure 5a, where a PDMS sheet (25 mm × 50 mm) is utilized as the freestanding triboelectric layer. During the first stage (Figure 5a-i), the freestanding PDMS sheet is brought into complete contact with the FTENG. Considering that the FTENG is initially uncharged, the triboelectric effect during contact will cause the PVDF fabric to be negatively charged while the nylon fabric to be positively charged. Simultaneously, this will render the part of the PDMS sheet that contacts the PVDF fabric to become positively charged, while the part that contacts the nylon fabric to become negative charged. When the PDMS sheet starts to move away from the FTENG as shown in the second stage (Figure 5a-ii), an electrical potential difference begins to develop that drives electrons from the PVDF electrode to the nylon electrode through an external circuit to counteract the generated triboelectric potential. As the separation between the PDMS sheet and the FTENG increases, an electrostatic equilibrium state is reached and the electron flow stops as shown in stage three (Figure 5a-iii). As the freestanding PDMS sheet is again driven towards the FTENG in stage four (Figure 5a-iv), the electrostatic equilibrium is broken and the electrons flow back from the nylon electrode to the PVDF electrode, thus reducing the number of induced charges. Finally, all the induced charges are neutralized when the PDMS sheet contacts the FTENG again. The FTENG performance was analysed by periodically contacting it with the freestanding PDMS sheet at an applied normal force of 3 N and frequency of 1 Hz, thus generating an alternating electrical output as shown in Figure 5b-i,ii. Under these conditions, the FTENG could generate an open circuit voltage (V_oc_) of about 14 V and short circuit current (I_sc_) up to about 28 nA.

### 3.3. WTENG Electrical Performance

The proposed WTENG is based on the previously discussed FTENG to enable implementation in wearable devices for harvesting mechanical energy from human motion and converting it into useful electrical energy. The structure and fabrication of the WTENG has been described in the previous section. Herein, we have investigated the non-deformation working mode of the WTENG as shown in Figure 6a. The WTENG is fixed on top of an acrylic block and periodically contacted by a freestanding PDMS sheet under normal applied force with varying magnitudes and frequencies. The electrical output of the WTENG when contacted with the same normal force of 5 N at different applied frequencies ranging from 1 to 5 Hz is shown in Figure 6b. It can be seen that the peak-to-peak voltage, current and power density increased from 3 V, 160 nA and 0.1 mW/m^2^ to 7 V, 700 nA and 0.85 mW/m^2^ as the applied frequency increased from 1 Hz to 5 Hz, respectively. The I_sc_ increases as more charges are transferred in a shorter time at higher frequencies. Since the internal resistance of the electrometer is not infinitely large, there will be charge leakage and the obtained values of the V_oc_ may be lower than those actually generated by the WTENG under different frequencies. Experimental analysis using calibrated resistances and capacitances could be utilized to figure out the exact value of the V_oc_ and internal capacitance of the WTENG based on input frequency [36]. We have also investigated the WTENG output under different normal applied forces ranging from 1 N to 5 N at a constant frequency of 1 Hz as shown in Figure 6c. It was observed that the peak-to-peak voltage, current and power density increased from 6 V, 80 nA and 0.12 mW/m^2^ to 14 V, 180 nA and 0.42 mW/m^2^ as the applied normal force increased from 1 to 5 N, respectively.

To characterize the practical applicability of the proposed WTENG as a power source, we measured the power density under different load resistances as shown in Figure 7a. A maximum instantaneous output power density of 0.065 mW/m^2^ was observed at a load resistance of 400 MΩ. Furthermore, we have also tested the current output of the WTENG when contacted with free standing triboelectric layers made of different materials as shown in Figure 7b. The peak-to-peak current of the WTENG when contacted with copper foil, latex, textile and PET (Polyethylene Terephthalate) under a normal applied force of 5 N (3 Hz) was 90, 95, 105 and 130 nA, respectively. This implies that PET is a relatively better triboelectric material for the WTENG. Furthermore, an output current of 105 nA for textile shows promise for applicability in wearable devices.

### 3.4. Applications of WTENG

Owing to its wearability, flexibility and ability to be triggered by any freestanding triboelectric layer, the proposed WTENG could be easily integrated into clothes, gloves, shoes and carpets. Besides the above tested materials like PDMS, latex, and PET; the freestanding triboelectric layer could also be wool, cotton, polymer, paper, the hand and so on. For example, a 5 cm^2^ WTENG placed on a table and tapped using the hand could generate a maximum peak-to-peak current of about 4 µA as shown in Figure 8a. The movement of hands, legs and arms represents important sources of mechanical energy provided by the human body. We have also integrated the same WTENG into clothing and can harvest energy by the motion of the arm as shown in Figure 8b. Here, the clothing material represents the freestanding triboelectric layer. The movement of the arm causes a change in the contact area between the WTENG and the clothing, resulting in a maximum peak-to-peak output current of about 0.6 µA. We have also demonstrated the feasibility of integrating the WTENG into floor mats. The WTENG was attached to the floor and the triboelectric effect resulting from stepping on it while walking generated a maximum peak-to-peak output current of about 2 µA as shown in Figure 8c. Based on a similar concept, the WTENG may also be attached to the bottom of shoes to harvest mechanical energy from walking or running.

## 4. Conclusions

In summary, we have demonstrated a woven structured TENG or WTENG comprising of PVDF nanofibers and commercial nylon cloth for biomechanical energy harvesting from human motion. The electricity is generated by a free-standing triboelectric layer that moves relative to the WTENG surface. A multinozzle centrifugal electrospinning process was employed for scalable manufacturing of PVDF nanofibers, which were also doped with MWCNT to increase the β-phase content and triboelectric charge generation. The proposed WTENG demonstrates feasibility for operating under contact with several freestanding triboelectric layers and can effectively harvest mechanical energy from a range of human motions including tapping, arm movement and footsteps with peak-to-peak current output reaching as high as 2 µA when the WTENG with dimensions of 5 cm^2^ is stepped on during walking. Consequently, the proposed WTENG shows promise as a power generator for wearable devices and can be extended to a range of other applications.

## Figures and Tables

**Figure 1 micromachines-10-00438-f001:**
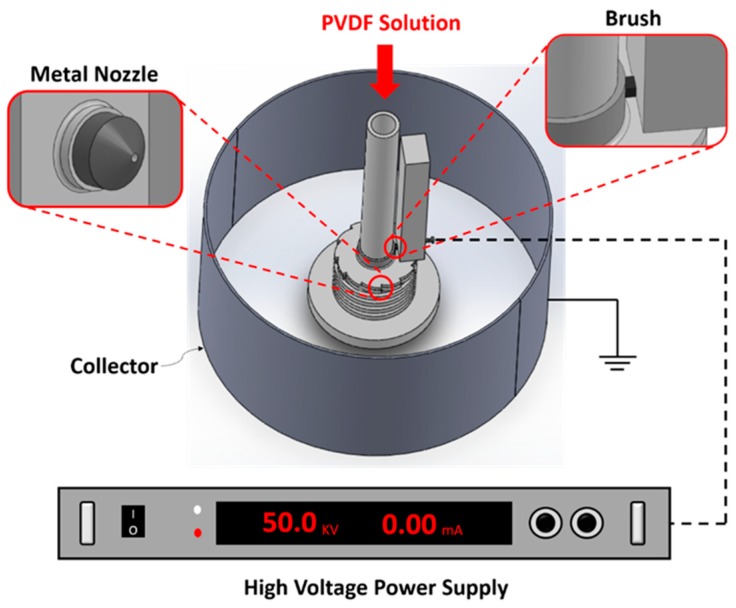
Schematic of the multinozzle far-field centrifugal electrospinning setup to fabricate the polyvinylidene fluoride (PVDF) nanofibers.

**Figure 2 micromachines-10-00438-f002:**
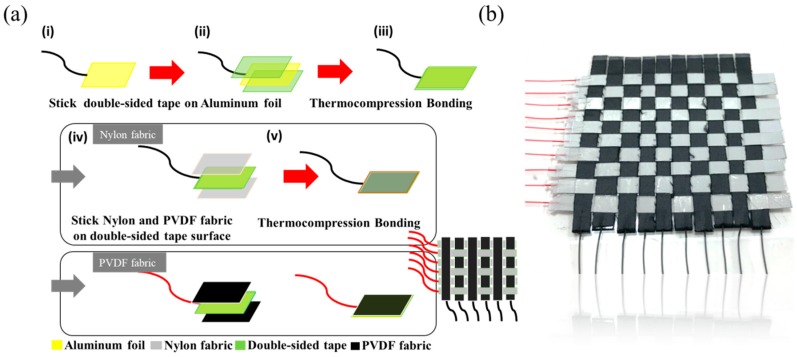
(**a**) Schematic illustration of the systematic protocol to fabricate the Woven Triboelectric Nanogenerator (WTENG). (**b**) An image of the 10 × 10 WTENG.

**Figure 3 micromachines-10-00438-f003:**
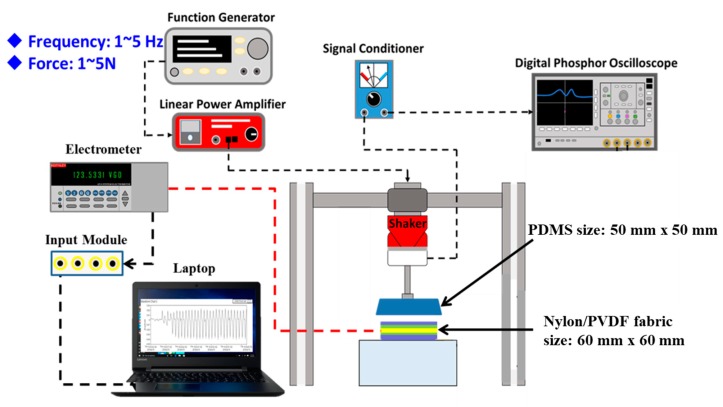
Schematic of the standard dynamic testing platform used to measure the electrical output of the WTENG under different applied forces and frequencies.

**Figure 4 micromachines-10-00438-f004:**
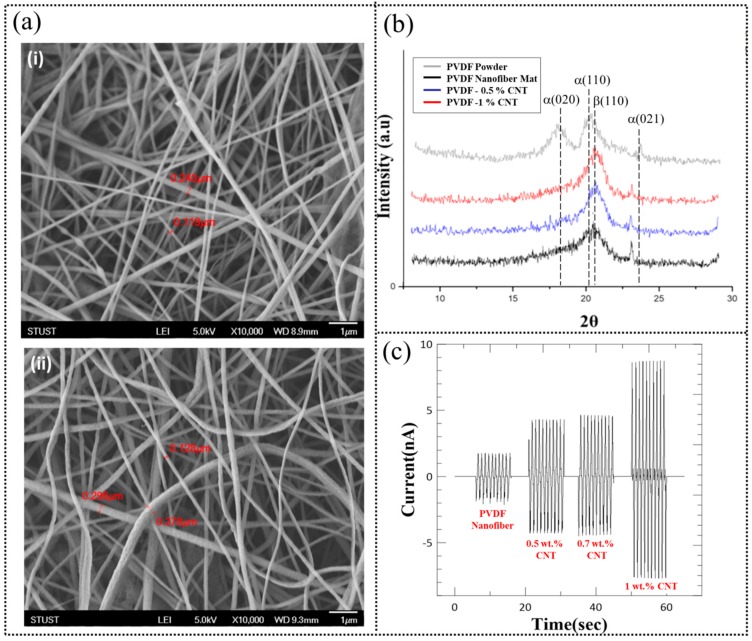
(**a**) FE-SEM images of (i) pure PVDF nanofibers and (ii) after doping with 1 wt. % MWCNT. (**b**) XRD spectra of PVDF powder and pure and MWCNT doped PVDF nanofiber mats obtained after centrifugal electrospinning. (**c**) Triboelectric current generation for pure and MWCNT doped PVDF nanofiber mats when contacted with a freestanding PDMS layer under an applied force and frequency of 3 N and 1 Hz, respectively.

**Figure 5 micromachines-10-00438-f005:**
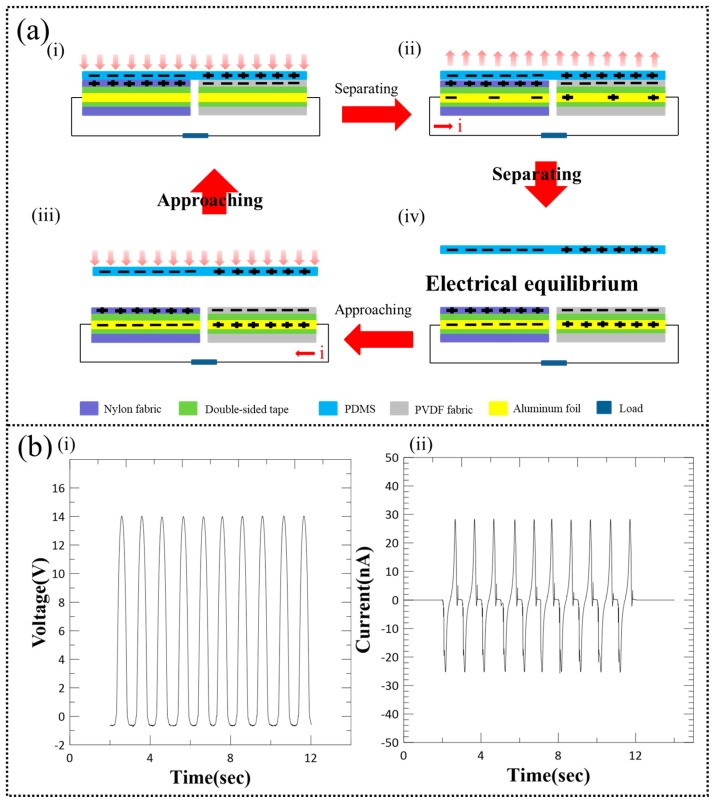
(**a**) Schematic of structure and working mechanism of freestanding TENG (FTENG) under vertical contact mode as the freestanding PDMS layer approaches contacts and separates from the FTENG. (**b**) Electrical performance of the FTENG: (i) Voltage output and (ii) current output.

**Figure 6 micromachines-10-00438-f006:**
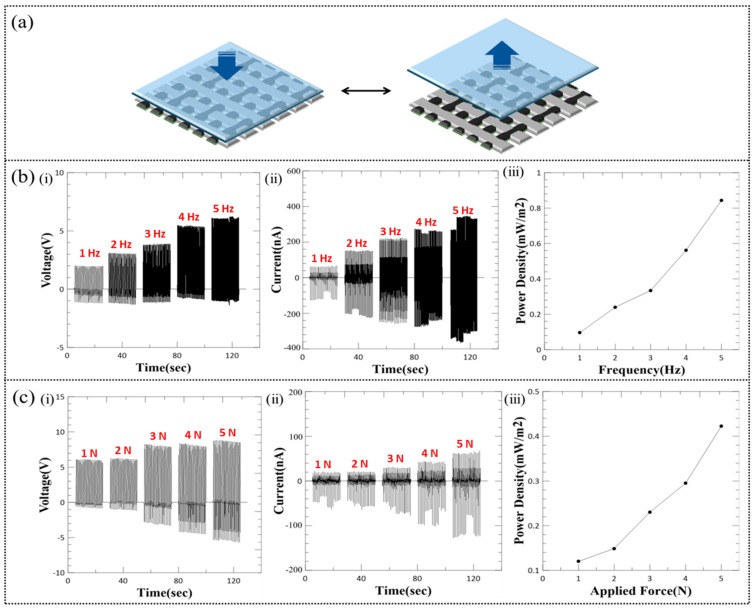
(**a**) Schematic of WTENG working mechanism under non-deformation mode where PDMS is chosen as the freestanding triboelectric layer. (**b**) The measured (i) voltage (ii) current and (iii) power density when the WTENG is contacted by the freestanding PDMS layer under varying (b) applied frequencies (1–5 Hz) and (**c**) contact forces (1–5 N).

**Figure 7 micromachines-10-00438-f007:**
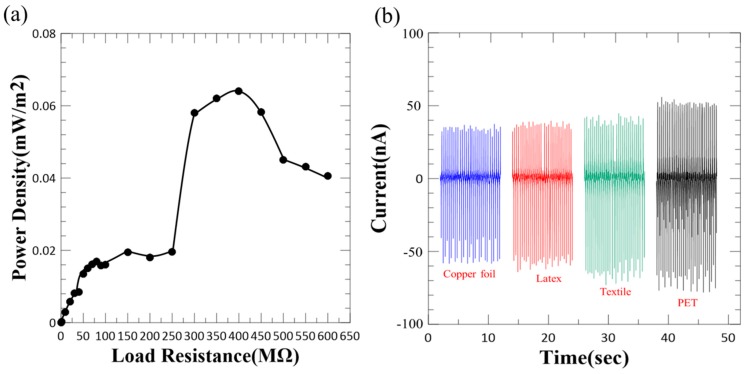
(**a**) The dependence of the WTENG power density on the resistance of the external load. (**b**) Current output of WTENG on contact with different freestanding triboelectric layers.

**Figure 8 micromachines-10-00438-f008:**
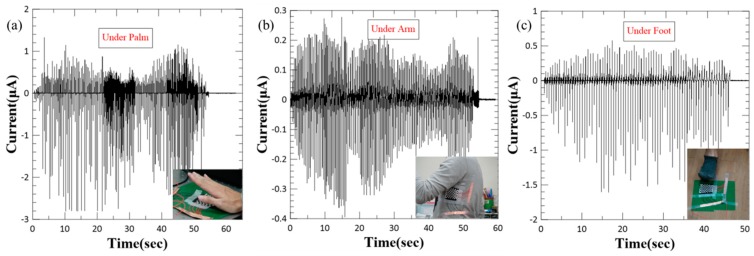
Applications of WTENG to harvest mechanical energy from human motion: (**a**) Tapping by hand. (**b**) Integrated into clothing to harvest energy from arm movement. (**c**) Stepping during walking.

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
