# Peer review of "Wearable Woven Triboelectric Nanogenerator Utilizing Electrospun PVDF Nanofibers for Mechanical Energy Harvesting"

_micromachines, 2019, doi:10.3390/mi10070438_

Round 1
Reviewer 1 Report
This article presents an unique structure of woven triboelectric nanogenerator with electrospun PVDF nanofibers. It presents certain amount of data with uniqueness and novelty. However, it did not clearly explain the relation of power enhancement with the doping of MWCNT and the phase of PVDF. The following are details of major and minor issues needed to be addressed before the publication.
For the analysis of PVDF, the author presented XRD data showing improvements of the ratio of beta-phase of PVDF. The analysis of PVDF’s phase only give the information about the presence of piezoelectricity, which may not be directly related with triboelectricity. Therefore, the other analysis should be afforded to explain the relation of PVDF’s phase and triboelectricity.
And the author mentioned the two role of MWCNT as a dopant such as improvements of crystallinity and electrical conductivity. The increase of beta-phase of PVDF does not mean the better crystallinity because alpha-phase PVDF is also crystalline form. To verify the crystalline, the other measurement such as DSC(differential scanning calorimetry) measurement need to be conducted. And it should be addressed how electrical conductivity can contribute triboelectric power enhancements and the reason why only 1 % is optimal value of MWCNT for mixture. Mostly, insulating tribo-material lose its surface charge if there exist conductive path.
Here are some minor issued as followed. the color legend of nylon in Figure 2 need to be changed because it seems very similar with the color of double-side tape. The value of electric field in the electrospinning process should be more informative to the Readers along with the applied bias.
Reviewer 2 Report
The authors reported a woven TENG with PVDF nanofibers using a new method. This manuscript can be accepted after minor revisions.
1, I highly suspect the statement "increase the degree of polarization (β-phase content) which results in enhanced triboelectric energy generation". Yes, increasing CNT composition do increase the β-phase content and the triboelectric charge generation can be enhanced, but the two things may not be related. Due to previous results, the polarized PVDF can either increase or decrease the triboelectric charge generation. (Nano Research 2014, 7(7): 990–997 DOI 10.1007/s12274-014-0461-8) For the non-aligned PVDF nanofibers as in the manuscript, its influence on the triboelectric charge generation is unsure. I wish the authors be cautious about their statement.
2, And for triboelectric charge generation, it is much better to show the charge output for comparison. Keithley 6514 can directly do that.
3, Fig 6b shows the voltage increase with the frequency which is a resistive behavior. Considering the Keithley 6514 is used for voltage measurement which is purely capacitive, there must be resistive leakage path between electrodes. The authors may discuss how to improve the fabrication so that this leakage path can be avoided.
Round 2
Reviewer 1 Report
The revised manuscript was thoroughly resolved major issues questioned in the comments. Now I believe it contains a novel data set and proper explanation for the publication.